# Development and Calibration of a Low-Cost, Piezoelectric Rainfall Sensor through Machine Learning

**DOI:** 10.3390/s22176638

**Published:** 2022-09-02

**Authors:** Andrea Antonini, Samantha Melani, Alessandro Mazza, Luca Baldini, Elisa Adirosi, Alberto Ortolani

**Affiliations:** 1Laboratory of Monitoring and Environmental Modelling for the Sustainable Development (LaMMA), 50019 Sesto F.no, FI, Italy or or or; 2Institute for the Bioeconony (IBE), National Research Council (CNR), 50019 Sesto F.no, FI, Italy; 3Institute of Atmospheric Sciences and Climate (ISAC), National Research Council (CNR), 00133 Rome, Italy

**Keywords:** low-cost acoustic disdrometer, rain gauge, machine learning, precipitation estimation

## Abstract

In situ measurements of precipitation are typically obtained by tipping bucket or weighing rain gauges or by disdrometers using different measurement principles. One of the most critical aspects of their operational use is the calibration, which requires the characterization of instrument responses both in laboratory and in real conditions. Another important issue with in situ measurements is the coverage. Dense networks are desirable, but the installation and maintenance costs can be unaffordable with most of the commercial conventional devices. This work presents the development of a prototype of an impact rain gauge based on a very low-cost piezoelectric sensor. The sensor was developed by assembling off-the-shelf and reused components following an easy prototyping approach; the calibration of the relationship between the different properties of the voltage signal, as sampled by the rain drop impact, and rainfall intensity was established using machine-learning methods. The comparison with 1-minute rainfall obtained by a co-located commercial disdrometer highlights the fairly good performance of the low-cost sensor in monitoring and characterizing rainfall events.

## 1. Introduction

The accuracy of measured precipitation data is of great importance for many scientific, commercial, and industrial applications, spanning from meteorological forecasting to hydrological design. Different types of instruments at the surface measure different properties related to rainfall flux. The conventional catching rain gauges (tipping bucket or weighing) are the most widely used instruments worldwide, and can measure an average quantity proportional to the time-integrated volume of an ensemble of raindrops [1]. Recently, commercially available non-catching devices (also known as disdrometers), have becoming more popular [2]. In addition to rainfall rate, disdrometers are able in real time to detect the size, velocity, and quantity of raindrops, thus providing information about microphysical processes of precipitation including the distribution of the number of precipitating drops as a function of their size, a quantity known as drop size distribution (DSD). The utility of different types of non-catching instruments has been demonstrated for several applications in hydrological sciences to describe characteristics of rainfall [3,4,5], radar meteorology [6,7], and validation of satellite estimates [8]. The major drawback is economic rather than technical. The high costs for purchase, installation and maintenance make the wide use of disdrometers over large areas difficult. Such devices can use different technologies and approaches: direct measurement of drop diameters and speeds observing their effects on the attenuation of laser beams, backscattering and Doppler effects on radiowaves, measurement of the impacts on a sensing surface that is converted into an electric signal by an electromechanical transducer, as in the early Joss–Waldvogel disdrometer [9] or by piezoelectric elements (e.g., [10]). The characteristics of commercially available piezoelectric rainfall sensors, along with the measurement principles both for the extraction of kinetic energy and DSD are described in [11]. Basically, their use for rainfall intensity estimation is physically based on the proportionality between the output signal strength and the volume of the falling drops. The aspects of piezoelectric sensing technologies and physical principles relevant for rain sensing are discussed in [12].

Piezoelectric sensors are generally less used than conventional catching rain gauges. One of their main issues is related to the calibration procedure. Most calibration processes involving piezoelectric rainfall sensors have been conducted so far in a controlled environment using rainfall simulators. However, when the measurements are taken in real conditions, with natural precipitation, additional errors can occur. This issue can be exacerbated if low-cost piezoelectric sensors are used, for example, because of higher system noise.

On the other hand, for many applications, especially in rainfall estimation, the availability of data from dense networks of sensors is crucial and justifies the research on low-cost sensors since the cost of professional ones could make the deployment of large dense networks unsustainable. Let us consider, for example, the intensification of very localized extreme precipitation events such as those analyzed in [13]; such precipitation events have highlighted some limitations of conventional measurement networks based on low representativeness of rain gauge distribution, which in some cases even fail to detect heavy precipitation events. New technologies have been developed for the deployment of dense sensor networks with improvements both in sensor size reduction and in sensor fusion and data processing [14,15]. For rainfall measurements, low-cost piezoelectric sensors can provide a viable solution. Moreover, the emergence of many machine-learning algorithms offers methods able to improve calibration also for many low-cost sensors for weather applications [16,17,18]. In some cases, low-cost weather sensors are calibrated by introducing, as ancillary predictors, other available environmental measurements. For example, ref. [19] proposes a method for calibrating air temperature measurements using co-located measurements of humidity, solar radiation, azimuth, and elevation, which are known to negatively affect the accuracy of low-cost air temperature sensors.

Aiming at pursuing both cost efficiency and a satisfactory performance of the device, in this work, a low-cost Piezoelectric Rainfall Sensor (PRS) was developed for monitoring and measuring instantaneous rain rate. It was developed from readily available materials, based on a low-cost piezoelectric transducer that senses the falling droplets by physical momentum transfer at the time of the impact of a droplet with the sensor. A novel approach based on machine learning (ML) was used for data calibration of the PRS prototype. Different ML methods were tested, thus allowing to investigate the relationship between the device electric response and the rainfall amount. Finally, the accuracy of the proposed calibration methods was assessed by comparing rainfall measurements with those obtained by means of co-located commercial instruments. The paper presents the development and the evaluation of the PRS and is organized as follows. Section 2 presents the setup of the instrument prototype, the description of signal and data processing including the adopted calibration methods testing different ML approaches. In Section 3, a discussion is provided about the resulting performances of the selected calibration methodologies. Finally, conclusions and future works are discussed in Section 4.

## 2. Materials and Methods

At the CNR (National Research Council of Italy) research area in Florence, Italy, with coordinates 43.818879 N, 11.201956 W and 50 m a.m.s.l, experimental equipment has been implemented and positioned on the roof of a building in an area with restricted access, at a height of about 18 m above ground level. The site is open on all sides and has no obstacles, above the horizon, represented by surrounding trees or buildings or other structures, that could prevent or distort the measurements of rainfall.

### 2.1. Instrument Prototype Setup and Signal Processing

The system developed in this work was designed and implemented following an easy prototyping approach to assess the effective feasibility and usability of the measuring instrument. For this reason, in this first development, the parts were put together using off-the-shelf and reused components. With reference to the architecture used in the development phase shown in Figure 1, it seems evident that computational resources are oversized with respect to real hardware and software needs; in fact, a recycled computer tower was used as an acquisition and processing system. However, the aim of this first prototype was development and experimentation concerning such a measurement system and assessment of the feasibility and effectiveness of the processing methods proposed in this work, i.e., to test processing procedures that are novel for this type of sensor, including machine-learning approaches for sensor calibration.

The sensitive element of the prototype is a piezoelectric sensor constituted by a unimorph piezo disc with an active ceramic piezoelectric layer (30 mm) put in contact with a brass substrate plate (50 mm). It is specifically designed to work at environmental condition temperatures in the range −20–70 °C. This piezoelectric transducer is glued to the underside of a glass plate that separates the sensor from atmospheric agents. The glass plate is slightly inclined (about 15°) to allow rainwater outflow from the sensing surface. The sensor housing is then arranged by a protective box containing the sensitive element and the wiring connections. Water particles impact the upper part of the glass plate and transmit the vibrations to the underlying piezoelectric sensor. In this way, a voltage signal is generated on the poles of the piezoelectric element. The voltage signal, as a function of the volume of the impacting drop [11], propagates trough a bipolar cable up to the electronic acquisition and control system consisting of an old Personal Computer (Compaq Evo D310). That system was equipped with an Integrated Analog Devices AD1981A AC’97 SoundMAX CODEC, an audio capturing device, used as an acquisition system directly connected to the piezoelectric transducer. The audio device is directly integrated in the motherboard of the Personal Computer and is managed through the PCI (Peripheral Component Interconnect) standard bus. In that audio acquisition system, the analog signal is sampled, quantized, and coded for conversion to digital. Specifically, the signal is sampled with a frequency of 8 kHz, and each sample is coded as a 16-bit floating point to ensure a good compromise between the available spectrum frequencies and the size of the generated file. Output data samples have a maximum value of +1.0 and a minimum value of −1.0, and are linearly correlated with input voltages to a good approximation [20]. Data produced by the acquisition system are then converted in WAVE audio file format for further processing. Each output file has an acquisition duration of 55 s and the sampling is repeated every minute, whereas 5 s are left for allowing the system to assemble and correctly save the file.

Figure 2 depicts an example of signal flow acquired by the PRS during a few seconds of a rainfall event. Note that, for visualisation purposes and only in this case, the signal amplitude was re-normalized with respect to the maximum recorded amplitude. The amplitude of the output samples can change depending on the software amplification factor; therefore such a factor must be set once and not changed during operations in order to obtain comparable measurements.

Once the acquisition phase is finished and the wave file is available, the signal-processing starts. A low-level signal intensity, which is comparable with the background noise of the system, is an indication of absence of precipitation. A threshold must be set as computed during no-rain periods to detect the instrumental and environmental noises that are related to the location where the system operates. The selected threshold corresponds to the maximum amplitude recorded during some selected no-rain periods. Any outlier, consisting of isolated values that far exceed the range of most recurring values should be identified and removed. Only the signal samples with an amplitude exceeding the environmental noise threshold provide information about precipitation occurrence. The time of the first sample with an amplitude exceeding the threshold is considered to be the start of a drop impact signal. Such a signal (Figure 2) is characterized by a maximum peak followed by a damped oscillation phase with the return to the equilibrium condition. From the calculation of the relative maxima of this trend, it is possible to reconstruct the envelope of the signal and therefore the damped trend. The end of the drop signal is defined as the time the drop envelope signal no longer exceeds the threshold, namely the first relative maximum below the noise threshold. It is known that the mean length of a drop signal is approximately 5–15 ms [21]. Therefore, a large time shifting window of 200 samples, corresponding to 25 ms, is used to identify the start and the end of the drop signal. The local maximum value is taken as the drop maximum amplitude, a quantity approximately dependent on the diameter of the drop through a power law [12]. In this time window, multiple drop impacts can occur altering a single drop signal duration. A method to identify these multiple impacts has been implemented based on the inversion of the envelope of the signal. If the impact of a drop occurs during the descendent relaxation time before the envelope is below the threshold, the signal undergoes a rapid rise and a new maximum can be detected. The inversion time of the envelope trend is used for the end of the first drop and the start of the second one. The processing phase leads to the computation of four output parameters referring to the 55 s acquisition period, namely the mean values of the drop signal duration (mean drop duration), the normalized (i.e., referring to a maximum value imposed by the audio acquisition device) maxima of drop signal amplitudes (mean drop amplitude), the number of identified drops (number of drops), and the overall sum of the absolute values of samples in the acquisition (sample sum). The data flow diagram of the processing applied to each acquisition time is shown in Figure 3.

For each 55 s wave file, the set of output parameters is computed and saved. An extensive dataset is then available for further processing and comparison with other measurement systems. Figure 4 shows an example of outputs provided by the implemented instrument, together with the spatially and temporally co-located rain rate measurements obtained from a professional disdrometer system, namely the OTT Parsivel 2 described in Section 2.2.

The correspondence between the outputs and the occurrence of precipitation seems evident, but the quantitative relationship between the values of each single output parameter with rainfall intensity is weak, although each recorded parameter is in relation with some precipitation characteristics. In fact, while the number of drops can be regarded as a proxy of the drop concentration number, assuming the existence of power law relations of the form $x=aD^{b}$ relating the raindrop diameter *D* to the amplitude or the the duration of drop signal yields that the sample sum, the mean drop amplitude, and the mean drop duration can be interpreted as different moments of the drop size distribution $M_{b}=a\int D^{b}N\left(D\right)dD$, whose order depends on the unknown exponents of the power laws. An approach that relates multiple output parameters of the sensor to the rain rate, which can also be regarded as a moment of the drop size distribution, is therefore justified.

Additional difficulties in finding an effective analytical relationship between the sensor output signal and rainfall intensity are due to low-cost assembly and quite basic signal processing. For instance, the sensing surface, despite its inclination, becomes wet during rain events, when the water layer can cause a dumping of the drop impact so that a part of kinetic energy cannot be transferred to the piezoelectric sensor, resulting in an attenuation of the signal and a possible alteration of the measured parameters. The water on the sensor also increases the noise level of the system, as can be noticed in Figure 4, by the trend of the sample sum quantity that increases following the beginning of the precipitation and remains at higher levels even when the precipitation stops. The environmental noise also contributes to errors in the retrieved parameters, as there is not a specific filter for this kind of noise. For example, we noticed that in the presence of strong wind, false alarms can be created or a variation of the sensor response. An analysis and attempts to resolve this type of unwanted effect would require the development of sophisticated signal-processing tools with development efforts that would risk canceling the low-cost aspect of the sensor. Based on these considerations, the proposed calibration procedure to relate impact measurements with rain rate estimation was carried out using machine-learning methods based on the set of output parameters as described above, so exploiting their connection with precipitation without any further processing.

### 2.2. Calibration with Machine-Learning Methods

The calibration of rainfall measurements from PRS is based on the assumption of the existence of a relationship between the sensor responses and the rainfall intensity. In most cases, the relationship is monotonous (i.e., as the voltage of the samples acquired by the sensor increases, the rain rate increases, according to either linear or power laws [10]). Following theoretical considerations expressed above and the experience gained carrying out this work, this monotonous relationship is not always respected and valid due to many causes, such as prolonged use of the instrument in the field, external and internal noise, etc.

In addition, the response of the sensor interface due to the drop impact is not easy to model, in particular after the beginning of the precipitation when the wet layer attenuates the impact of the falling drops as discussed above. Irregular and unexpected sensor responses are more frequent when low-cost sensors are used [17,22].

We applied a software-based machine-learning method in the calibration of the PRS, as shown in studies in the literature demonstrating that such approaches can effectively improve the performances of low-cost environmental sensors [23]. The availability of a co-located laser disdrometer instrument allowed a direct comparison between the reference measured instantaneous rain rate and the response of the sensor under study. Such reference data have been used in the training and test steps of the machine-learning model setup. The laser disdrometer considered in the study is a PARSIVEL (particle size and velocity) disdrometer, second generation, manufactured by OTT GmbH (Kempten, Germany). It has an optical sensor that produces a horizontal sheet of light that is focused on a single photodiode. Passing through the light sheet, particles partially blocked this light sheet, causing a short reduction in voltage in a clear sky (equal to 5 V). The amplitude of the reduction is proportional to the dimension of the drops while the fall velocity depends on the duration of the reduction. The software of the manufacturer provides the number of drops in 32 diameter size and 32 fall velocity categories, with variable widths. The particle size ranges from 0.062 to 24.5 mm, while the fall velocity ranges from 0.05 to 20.8 ms${}^{-1}$. However, the first two size categories, which correspond to sizes less than 0.2 mm, have been left empty due to the low signal to noise. Knowing this information, the drop size distribution can be obtained and the rainfall rate can be straightforwardly computed [8]. The PARSIVEL disdrometer rain rate measurements are averaged over a minute of acquisition. For the training phase of the ML approach, data relative to the month of April 2019 were selected. The total amount of rainfall during April 2019 was 66.8 mm, with eight rainy days. The full month dataset acquired by both the low-cost sensor and the reference disdrometer was randomly split into two parts: 75% used for the training phase and 25% for the test phase.

Most machine-learning methods are based on data analysis and empirical choice of the best regression/classification method. With this rationale, some different machine-learning algorithms were tested for the sensor-calibration process. The selected methods (implemented in the Scikit-learn framework [24]), together with the results on the analysis on the Mean Absolute Errors (MAEs) obtained during the training and test phases are shown in Table 1.

As expected, the joint use of the sensor output quantities and the measurements of a reference equipment for training and test showed that the instrument does not provide signal properties that are correlated with the precipitation rate. In fact, the results of machine-learning methods based on linear models (i.e., Linear Regression, Partial Least Squares Regression) show poorer performances. Moreover a Support Vector Regression algorithm was applied using radial basis as the Kernel function, i.e., an operator to remap the original sensor output signals to a new space of vectors that are linearly combined to obtain the output. The performances of the Support Vector Regression are very similar to linear models, demonstrating the difficulty of the task of linearizing the relationship between the parameters provided by the sensor and the precipitation intensity. The Multi-Layer Perceptron Regressor (MLPR) neural network algorithm, very suitable in cases of non-linear relations, was tested with 500 passes over the training data; the results of this method on MAE both in training and in test phases are comparable or even worse than the methods described before.

Even though these preliminary tests were carried out on all the listed methods of Table 1, the final analyses will focus only on those that showed the best performances in the training and/or testing phase: Decision Tree Regression (DTR), Random Forest (RF), and K-Nearest Neighbors Regression (KNNR). The first two methods follow quite similar approaches, as Random Forest is an extension of DTR that combines multiple decision trees, with better results for the RF and DTR in the testing and training phases, respectively. The KNNR method, implemented to consider the five nearest neighbors, showed instead the worst value of MAE in the training phase, but the best one in the testing phase.

Considering these results, the selected methods were applied to a dataset completely independent from that used for the training and testing of the ML. For this dataset, reference disdrometer data were available as well. The results will be shown and discussed in the next sections.

## 3. Results

The most promising ML approaches in terms of MAE statistics have been applied to significantly extended and independent datasets for assessment of their performances. Two months of data (May and July 2019) were used, characterized by a monthly cumulated rainfall of 111 and 94 mm, respectively. In May, precipitation was distributed over 11 rainy days, while in July rainfall was concentrated only in 3 days. The selected dataset refers to different precipitation events with very different intensities, some of them with peaks of high rain rate and duration of a few minutes, while other events have a reduced rainfall rate but a duration of several hours; thus the dataset, as shown in the next figures, includes different precipitation types, from very light to heavy precipitation. The device outputs were temporally matched with the disdrometer data in order to validate the 1-minute rain rate as estimated by the selected ML algorithm. The rainfall outputs from DTR, RF and KNNR approaches are thus compared with the disdrometer-estimated precipitation. The results are shown in detail for some selected days of the datasets in order to evaluate the ML algorithm performance in detecting and characterizing the intensity and duration of the precipitating events under study.

Figure 5 refers to a day of light rain (i.e., 4.6 mm of cumulated daily rainfall). The rainfall event ended in just a few hours with an intensity peak of about 25 mm h${}^{-1}$ just before 12 UTC. All the ML methods correctly detect the rainfall event and its evolution in time. The four peaks are well reconstructed in number, intensity and temporal evolution with DTR and RF methods, while KNNR underestimates the value of all the rainfall peaks, especially the maximum one. This behaviour of KNNR is partially in contrast with that shown in a different case (Figure 6), where a rain rate peak value of more than 60 mm h${}^{-1}$, as registered by the Parsivel disdrometer, is well reconstructed by KNNR, while the estimated values provided by DTR are more than 140 mm h${}^{-1}$ and RF slightly overestimates the first peak. This rainfall case is a two-day event with cumulated rainfall values of 10.2 mm and 19.2 mm (18 May and 19 May, respectively). In the first day, rainfall is concentrated in a few minutes (around 13:00 UTC), while a light rainfall is more uniformly distributed over a few hours in the second day. For this case study, the best overall performance is provided by RF, as it correctly reconstructs in time and intensity the different rainfall patterns distributed in the two days’ events.

A further two days of events with cumulated daily rainfall values of 7.6 and 17.4 mm (28 and 29 July, respectively) are depicted in Figure 7. The peak of rainfall estimated by all the methods on the second day (28 May) without any real correspondence with the measurements of the disdrometer sensor stands out. This peak occurred during the period of maximum intensity of the precipitating event, but nevertheless it was largely overestimated by all the methods, with KNNR showing the smallest error. The precipitation patterns of the first day (27 May) were well detected by all the methods, with the DTR showing the larger overestimation of the peak value.

Finally, Figure 8 shows a case relative to 15 and 16 July in which only the first day was rainy (9.6 mm). Light precipitation falls almost intermittently throughout the first day, with a stop between 09 and 12 UTC. All the methods provided false alarms of very light rain before 12 UTC of 15 July and during the first hours of 16 July, although with very small precipitation values. Also in this case, the KNNR proves to be the method that shows better performances in terms of both the estimated rainfall intensities and the temporal evolution of the precipitation pattern. The DTR seems to provide the worst results, deviating in a more consistent way from the disdrometer measurements.

Figure 9 show the scatter plots of the rain rate estimated from the impact sensor and the reference disdrometer for the three different methods.

To quantify the ML approach performance, MAE values were computed for all the selected methods for the whole validation dataset (May and July 2019). The results are shown in Table 2. KNNR and RF methods provide the best performances, and in any case not too far from the DTR ones. MAE values, very close to 0.1 mm${}^{-1}$, are comparable with the minimum resolution (i.e., 0.2 mm h ${}^{-1}$) of the most common tipping bucket rain gauges.

## 4. Conclusions and Future Works

In this work, the development of a low-cost piezoelectric rainfall sensor (PRS) and its calibration to estimate rainfall intensity has been described. The results in terms of precipitation patterns compared to those of a reference instrument have shown that the application of machine-learning methods for the sensor calibration can provide performance comparable to that of professional instruments (let us refer, for example, to the numerical results of the comparisons between three widely used laser disdrometers shown in [25]).

A discussion about some selected machine-learning algorithms has been provided, highlighting the methods that are slightly better in the shown cases.

For the implementation of the sensor prototype, some re-used materials were employed. The housing of the sensor was made of wood properly treated to be protected from atmospheric agents. An old desktop Personal Computer, no longer usable even for very basic office works, configured with open source OS and applications, was used as the signal conditioning and acquisition system and to implement the processing. The most expensive part of the entire system was the development of the signal-processing software and the critical task of calibration that was performed using machine-learning algorithms. Clearly this part of the development is carried out only at the first implementation, and can be reused in the following tests replicating the prototype. In this way, the implementation costs can be drastically reduced.

The piezoelectric sensor is practically inexpensive and its housing could be easily realized with a 3D printer (let us consider EUR 20 for both the sensor and the protecting box). The control and processing system should be realized with a mini PC, for example a Raspberry PI 3 (EUR 50) and a USB sound card could be used as acquisition system (EUR 20), for a total of EUR 90 of hardware. Target costs for the production of a series of PRS need to include those related to the critical calibration phase that can be recovered depending on the produced sensors.

However, some further steps would require a system-engineering process involving a redesign of the system architecture in terms of the sensing surface dimensions, choice of the piezoelectric sensor, etc., an optimization of the processing software and a standardization of the calibration procedure that should be performed on site by a customer.

Although the sensor has been tested over a period of two months with a certain number of precipitation events even of different intensities, a structured validation work should be properly designed and carried out, especially in different seasons and on different types (convective/stratiform) of precipitation.

Given the clear need for further technical and scientific development including research work to investigate the use of the sensor for a direct measurement of the rainfall particle sizes, the prospect of operational use of this type of sensor seems equally evident. For the purpose of using it as a disdrometer, some specific tests should also be conducted to find correlations between the environmental conditions (wind, humidity, etc.) and the acquired measurements, with the aim to correct them.

Some important benefits could be related to the introduction of an Internet of Things (IoT) paradigm, supporting the implementation of distributed and interconnected sensor networks and citizen science applications.

## Figures and Tables

**Figure 1 sensors-22-06638-f001:**
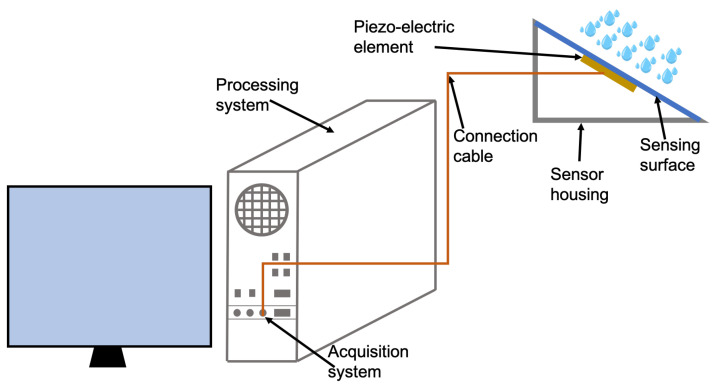
Prototype architecture of the PRS.

**Figure 2 sensors-22-06638-f002:**
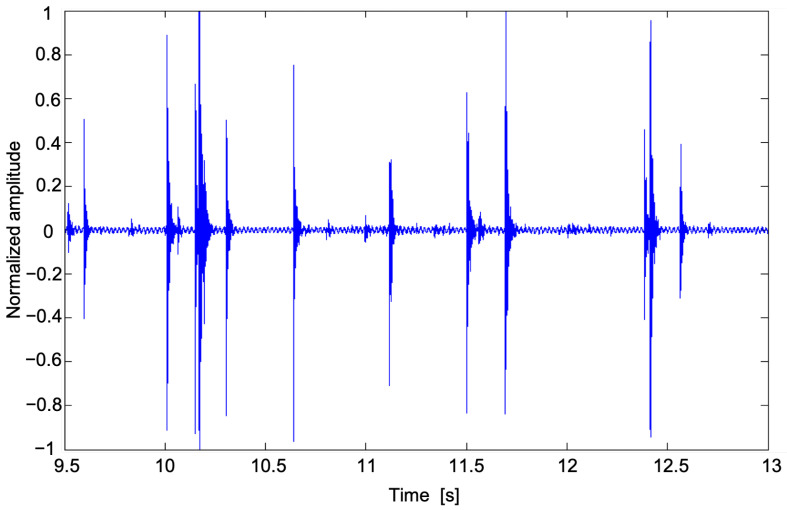
The sensor signal (normalized amplitude) in time domain.

**Figure 3 sensors-22-06638-f003:**
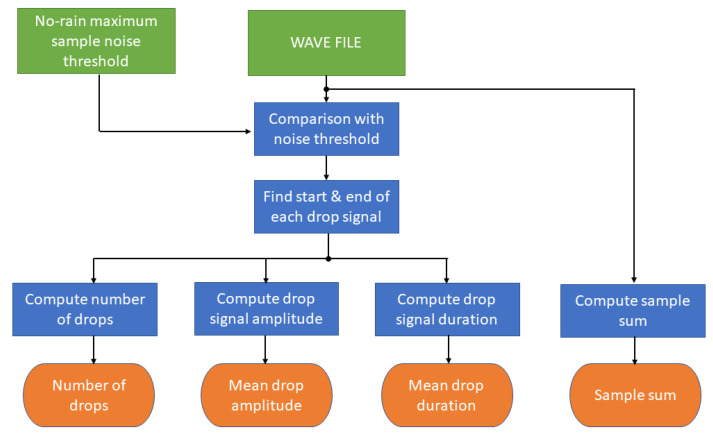
Block diagram of the processing for each 55 s wave file (see text for description of the blocks).

**Figure 4 sensors-22-06638-f004:**
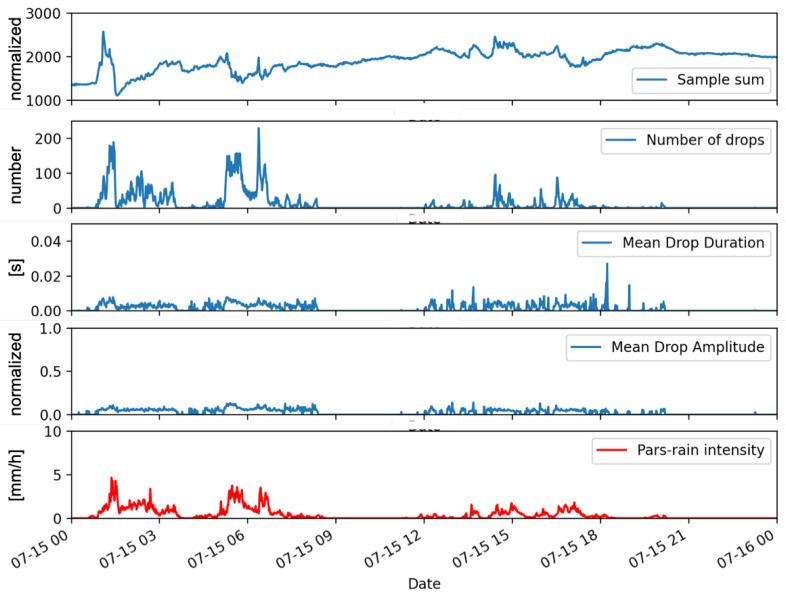
Outputs from the impact disdrometer quantities (graphs in blue) as compared with the 1-minute averaged rainfall intensity obtained by a commercial disdrometer (graph in red).

**Figure 5 sensors-22-06638-f005:**
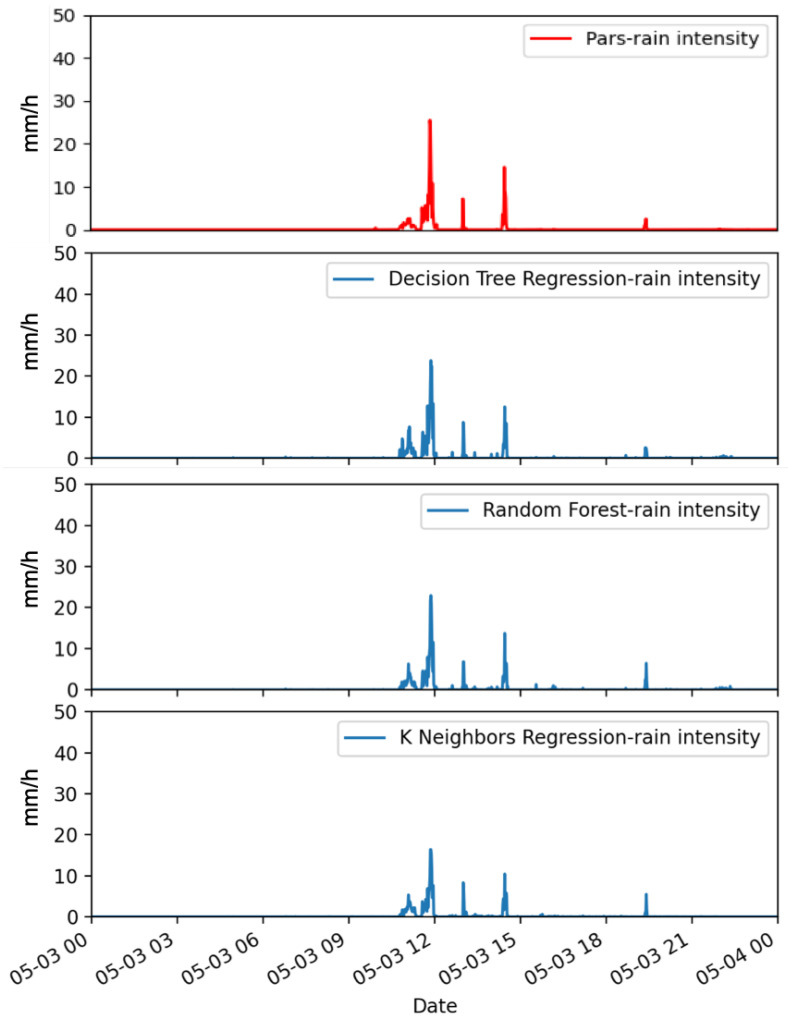
Instantaneous rainfall rate as measured by a co-located reference laser disdrometer (on top) and as estimated by the ML (DTR, RF, KNNR) approaches (from top to bottom, respectively). Data are relative to 3 May 2019.

**Figure 6 sensors-22-06638-f006:**
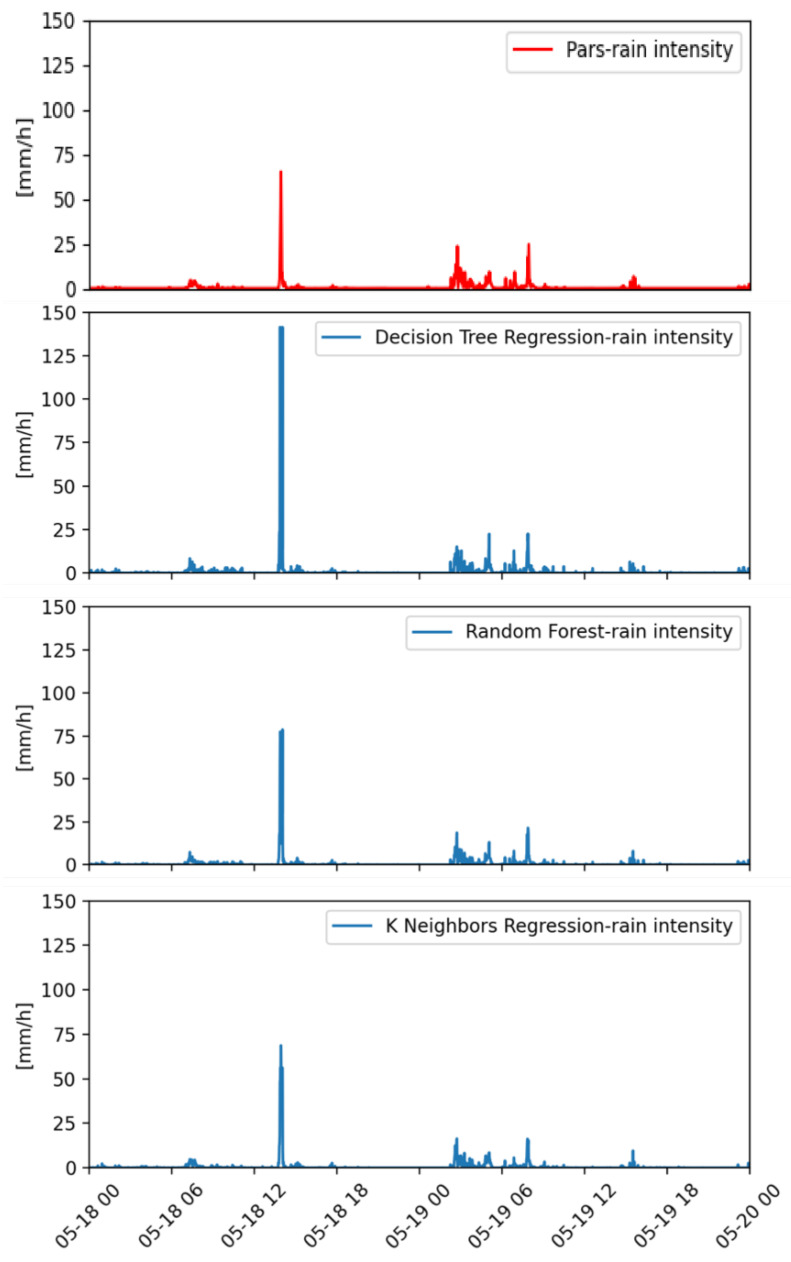
Instantaneous rainfall rate as measured by a co-located reference laser disdrometer (on top) and as estimated by the ML (DTR, RF, KNNR) approaches (from top to bottom, respectively). Data are relative to 18 and 19 May 2019.

**Figure 7 sensors-22-06638-f007:**
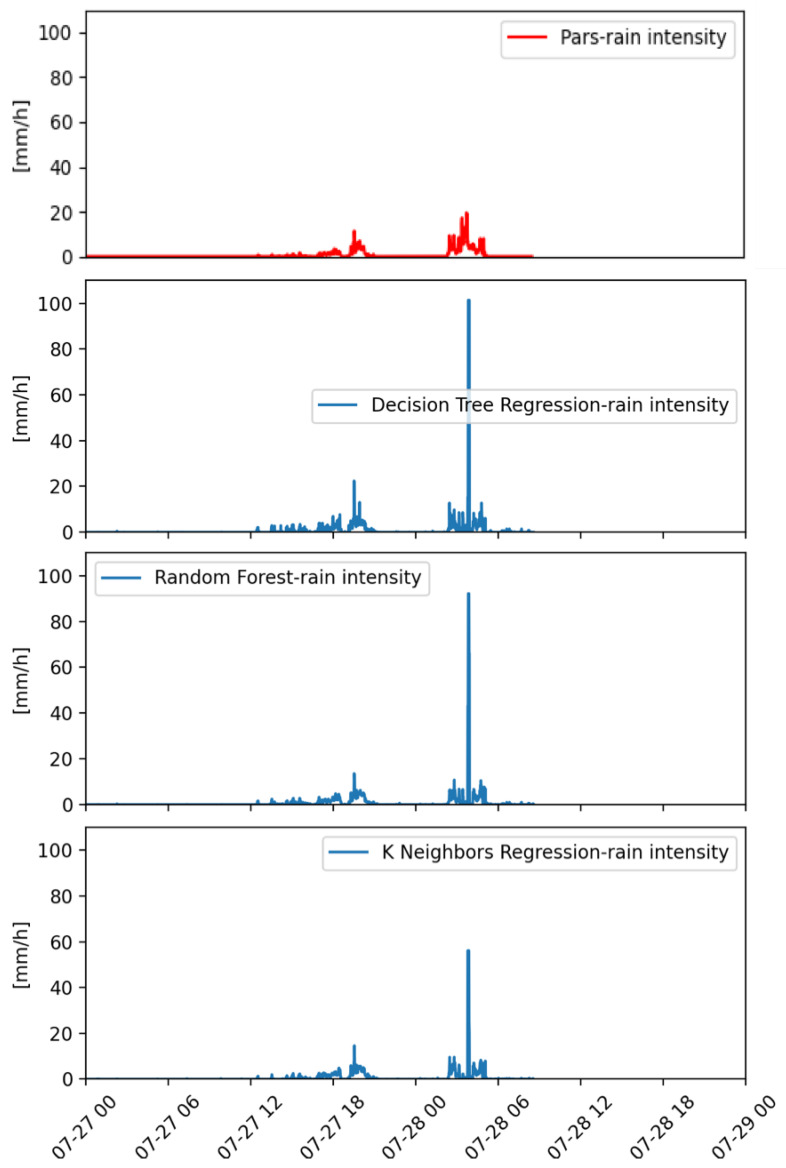
Instantaneous rainfall rate as measured by a co-located reference laser disdrometer (on top) and as estimated by the ML (DTR, RF, KNNR) approaches (from top to bottom, respectively). Data are relative to 27 and 28 July 2019.

**Figure 8 sensors-22-06638-f008:**
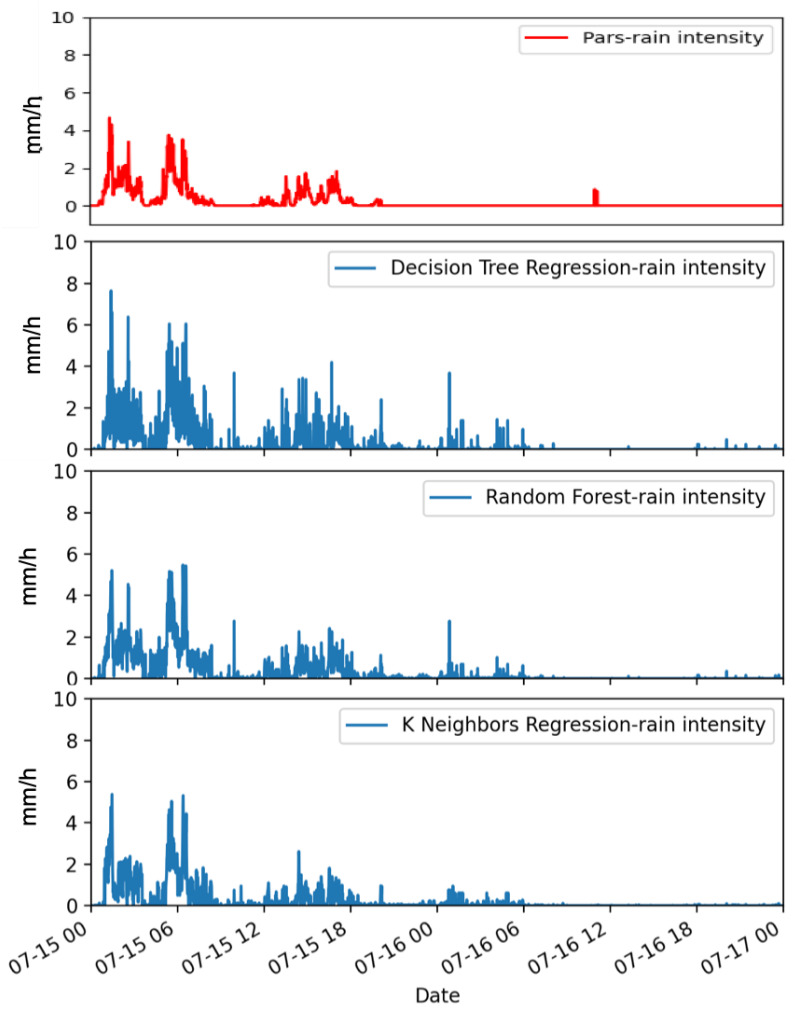
Instantaneous rainfall rate as measured by a co-located reference laser disdrometer (on top) and as estimated by the ML (DTR, RF, KNNR) approaches (from top to bottom, respectively). Data are relative to 15 and 16 July 2019.

**Figure 9 sensors-22-06638-f009:**
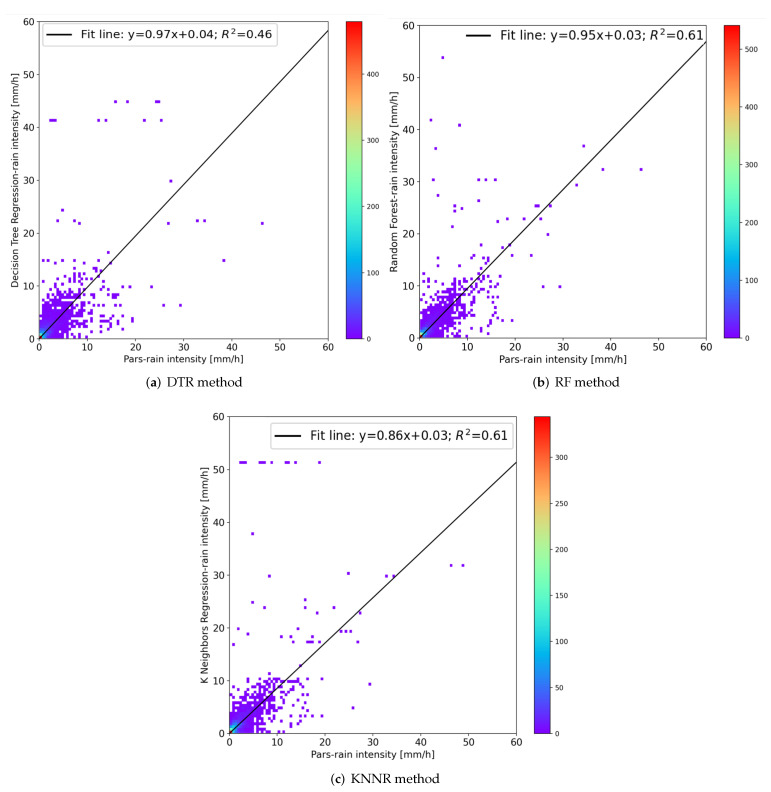
Comparison of the 1-min rainfall intensities estimated by the PRS with the reference laser disdrometer for all the two-month period (May and July).

**Table 1 sensors-22-06638-t001:** Comparison between different machine-learning methods applied to the dataset (April 2019). MAE train and test are the Mean Absolute Errors obtained for the training and test dataset, respectively.

Method	MAE Train (mm/h)	MAE Test (mm/h)
Linear Regression	0.15	0.13
Partial Least Squares Regression	0.17	0.15
Support Vector Regression	0.14	0.13
Decision Tree Regression	<0.01	0.06
Random Forest	0.02	0.05
K Neighbors Regression	0.05	0.04
Neural Network MLPR	0.17	0.15

**Table 2 sensors-22-06638-t002:** Comparison between three different machine-learning methods applied to the analysed data. MAE (Mean Absolute Errors) obtained for May and July were considered.

Method	MAE in May and July
Decision Tree Regression	0.11
Random Forest	0.08
K Neighbors Regression	0.08

## Data Availability

The data of this study can be distributed upon request to the corresponding author.

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
