# Peer review of "Development and Calibration of a Low-Cost, Piezoelectric Rainfall Sensor through Machine Learning"

_sensors, 2022, doi:10.3390/s22176638_

Round 1

Reviewer 1 Report

The authors explore the use of a piezoelectric rainfall sensor, developed at minimal cost, to estimate rainfall intensity.  A good background and description of the sensor is followed by experiments using various statistical methods.  The authors also nicely illustrate the low cost of this rainfall estimation method in the conclusions.  Overall, this is a well-written and organized paper.  I only have minor comments, which I detail below.  

Minor Comments:

1)  My most significant comment is about sample size.  While this is a quick experiment that does show promise given the results from Table 1 and deserves publication, it is unlikely the authors have sampled a good range of events for calibration.  Readers should be cautioned about this, and the authors may wish to consider increasing the number of events for calibration in future work to improve the sampling of different precipitation regimes (convective vs. stratiform).  

2)  Readers may wonder about the location where this work was done.  A brief description of the site where this experiment took place is recommended.

3)  Line 230:  Are these really poor performances?  They are certainly poorer relative to the other methods in Table 1.  I would change poor to poorer unless the authors have other evidence that shows no statistical skill.

4)  Line 259:  I think about precipitation regimes in terms of convective and stratiform, and just because rain occurred over fewer days does not suggest different precipitation regimes.  In fact, Figures 7 and 8 might suggest different regimes in July alone (lighter precipitation in Figure 8 vs. Figure 7).  Minor rewording here is recommended.

Author Response

REPLY TO REVIEWER n.1

Dear reviewer,

Thank you very much for your comments concerning our manuscript entitled “Development and calibration of a low-cost, piezoelectric rainfall sensor through Machine Learning”. We have examined all your comments and suggestions carefully and made some revisions accordingly (please check the highlighted in blue in the new version). The following are our explanations for your questions or suggestions in your reviews.

R1Q1: My most significant comment is about sample size.  While this is a quick experiment that does show promise given the results from Table 1 and deserves publication, it is unlikely the authors have sampled a good range of events for calibration.  Readers should be cautioned about this, and the authors may wish to consider increasing the number of events for calibration in future work to improve the sampling of different precipitation regimes (convective vs. stratiform).  

Answer1: Thanks to the reviewer for this comment. In fact, the intent of the article is not to exhaustively show the sensor's performance in all precipitation regimes, but to describe its development process and the innovative method adopted for its calibration. Actually, the two months of data shown seem to be related to different precipitation regimes, but we are certainly far from a validation campaign on an extended dataset for different seasons and types of precipitation. We added a sentence in the conclusions section to clearly specify this aspect.

R1Q2:  Readers may wonder about the location where this work was done.  A brief description of the site where this experiment took place is recommended.

Answer2: A brief description about the site used for the experiments has been added at the beginning of in Section 2

R1Q3:  Line 230:  Are these really poor performances?  They are certainly poorer relative to the other methods in Table 1.  I would change poor to poorer unless the authors have other evidence that shows no statistical skill.

Answer3: You are absolutely right (it was a typing error). The text has been changed accordingly. Many thanks

R1Q4:  Line 259:  I think about precipitation regimes in terms of convective and stratiform, and just because rain occurred over fewer days does not suggest different precipitation regimes.  In fact, Figures 7 and 8 might suggest different regimes in July alone (lighter precipitation in Figure 8 vs. Figure 7).  Minor rewording here is recommended.

Answer4: Sorry; the text was misleading w.r.t. what we wanted to communicate. The fact that in the months of May and July 2019 almost the same amount of rain fell but that the rainy days were significantly less in July (3 versus 11) suggests that on average the precipitation intensity during the month of July was greater. We wanted to link this sentence to the fact that the results shown refer to different precipitation events with very different intensity values. For example, the event of figure 6 has a peak of intensity greater than 50 mm / h lasting a few minutes, while the event shown in figure 8 had a reduced rainfall rate but lasted several hours.

Thanks for the report. We rewrote the sentence.

Reviewer 2 Report

Manuscript ID: sensors-1854422  
Title: Development and calibration of a low-cost, piezoelectric rainfall sensor through Machine Learning
OVERVIEW
The manuscript presents the development of a prototype of an impact rain gauge based on a very low-cost piezoelectric sensor developed by assembling off-the-shelf and reused components.
GENERAL COMMENTS
The subject matter is actual, interesting and within the scope of the Journal Sensors.
Some changes must be made in the manuscript to comply with the journal template.
The English spelling and grammar are very good.
My major concern is that the author does not give a detailed description of the “sensitive element” used in the experiments and there are some installation aspects that may jeopardize the results, e.g. the inclination of the sensor and 55 seconds of observation per minute.
As for the rest, I have a few suggestions. Please read the specific comments.
In conclusion, I believe this manuscript is interesting and worthy of publication after major changes.
SPECIFIC COMMENTS
Line 77: Please follow the journal template and include a section “Materials and methods”. Sections 2 and 3 may be sub-sections.
Line 90: Please describe the material and the exact reference of the component used as a “sensitive element”.   
Line 92: The inclination of 15 degrees affects the measured values. There is any correction to remove the effect of the inclination from the measured data. Please explain.
 Line 107: The rainfall is recorded for 55 seconds. In the remaining 5 seconds where the data is assembled and saved there is any measurement? There is an error of 55/60 = 8.33% because of this option. Can this be avoided by parallel processing? Please explain.
Line 177: reads “The analysis and attempts to resolve this type of unwanted effects would require the development of sophisticated signal processing tools”. Did the authors try to introduce a windshield in the perimeter of the sensor, so the rainfall drops would hit the sensor vertically? As the sensor inclines 15 degrees, the wind direction has a direct effect on the angle of the raindrops' impact on the sensitive element and consequently on the kinetic energy of the raindrops' impact and the sensor signal.  
Line 184: the actual section “Calibration with machine learning methods” should also be a sub-section of “Materials and methods”.

Author Response

REPLY TO REVIEWER n.2

Dear reviewer,

Thank you very much for your comments concerning our manuscript entitled “Development and calibration of a low-cost, piezoelectric rainfall sensor through Machine Learning”. We have examined all your comments and suggestions carefully and made some revisions accordingly (please check the highlighted in blue in the new version). The following are our explanations regarding your questions, comments and suggestions.

R2Q1: Line 77: Please follow the journal template and include a section “Materials and methods”. Sections 2 and 3 may be sub-sections.

Answer1: The section “Material and methods” has been introduced as section 2 and consequently we have subsections “2.1. Instrument prototype setup and signal processing” and “2.2. Calibration with machine learning methods”

R2Q2: Line 90: Please describe the material and the exact reference of the component used as a “sensitive element”.  

Answer2: For a reference to the piezoelectric transducer used in this work please refer to: https://www.alibaba.com/product-detail/Piezo-Disc-Piezo-Disc-8-50mm_60093770186.html?spm=a2700.7724857.0.0.773f2ffb7a3SMt&s=p

As this type of site is only a sales site, we did not include it in the bibliography.

The piezoelectric transducer serial number is: KS-502T10A, it is constituted by an unimorph piezo disc composed by one active piezoelectric layer and one non-active layer. The substrate plate is in brass (50mm) with the and the ceramic piezo element (30mm). It can work at every environmental condition temperature in the range -20°C,70°C.

We added such further detail in the text.

R2Q3: Line 92: The inclination of 15 degrees affects the measured values. There is any correction to remove the effect of the inclination from the measured data. Please explain.

Answer3: The inclination is necessary to let the water flow off the impact surface. We did not make any correction for the inclination as well as there are no corrections for other factors such as wind and other atmospheric parameters, the position of the sensor with respect to the point of impact of the drop, or other factors, some of which detailed in the text. However, the correction of these effects should be in large part implicitly accounted by the calibration procedure through machine learning methods. This is clarified in the text at lines 200-206.

R2Q4: Line 107: The rainfall is recorded for 55 seconds. In the remaining 5 seconds where the data is assembled and saved there is any measurement? There is an error of 55/60 = 8.33% because of this option. Can this be avoided by parallel processing? Please explain.

Answer4: The main concern about the 5 seconds left for the data file finalization was due to trying not to burden with too many writes on the hard drive. In fact, also during the acquisition process from the sensor (using the “arecord” linux command) data are written on the hard drive and we wanted to avoid double writes (on a rather old hard drive). About the effects of sampling 55 seconds instead of 60, note taht the reference disdrometer measurements are averaged over one minute. As the calibration of the sensor (during the training and test phases of the ML experiments) is made with the reference sensor, at the end of the process also the measurements taken by the sensor provide one minute averaged precipitation. We could argue that a non negligible error could arise if the precipitation rate in the non-acquisition 5-seconds period changes abruptly w.r.t. the precedent 55-seconds of recording, but it is unlikely to happen and in addition the average factor acts as a smoothing of this error component.

Thanks to the comment of the reviewer, we realized that it was not written that the measurements of the reference disdrometer were averaged over one minute of acquisition, so we have introduced it in the text (in the section where the disdrometer is described).

R2Q5: Line 177: reads “The analysis and attempts to resolve this type of unwanted effects would require the development of sophisticated signal processing tools”. Did the authors try to introduce a windshield in the perimeter of the sensor, so the rainfall drops would hit the sensor vertically? As the sensor inclines 15 degrees, the wind direction has a direct effect on the angle of the raindrops' impact on the sensitive element and consequently on the kinetic energy of the raindrops' impact and the sensor signal.

Answer5:

Windshields and other precautions are suggested by WMO requirements to get any type of rainfall and other atmospheric measurements. However, these are not always adopted due to operational condition of installations.

In the specific case of this work, as the experiment was conducted on the roof of a building and the use of wind protection screens was not implementable (due to the risk that, in case of strong winds, they could fall from the roof of the building and cause damage to people and or things).

A specific study should be interesting to assess eventual dependences of the acquired measurements from the environmental conditions. This type of investigation should be particularly important if measurements are used to characterize drop size distribution.

About this aspect a specific sentence has been added in the “Conclusion and future work” section.

R2Q6: Line 184: the actual section “Calibration with machine learning methods” should also be a sub-section of “Materials and methods”.

Answer6: The section “Material and methods” has been introduced as section 2 and we have transformed into subsections “2.1. Instrument prototype setup and signal processing” and “2.2. Calibration with machine learning methods”

Reviewer 3 Report

This article is written clearly and well organized. The introduction given and the background given are reasonable given the premise of the article. Figures and tables are comprehensive and helpful to know in detail about the article. Apart from this, the author needs to check for the comments given below.

a.      Many typographical problems must be fixed before publication (th- to be represented in the superscript likewise fewer alphabet which needs to specify in the subscript also needs to be specified properly).

b.      The author has to provide a few lines about the disdrometer in the introductory section, and they also need to address the benefits and drawbacks of using a disdrometer.

c.      The author must include the application of the novel approach that is based on machine learning (ML).

d.     Since commercially accessible non-catching devices (also called as disdrometers) that employ various technologies to gather not just rainfall rate but also Drop Size Distribution (DSD) have gained in popularity. The author need to  discuss DSD in a few sentences?

e.      A low-cost piezoelectric rainfall sensor (PRS) was designed for the purpose of monitoring and detecting the instantaneous rainfall rate. This was done with the goal of achieving both cost efficiency and good gadget performance. Could the author provide much more details about the PRS?

f.      The PCI (Peripheral Component Interconnect) card, where the analog signal is sampled, quantized, and coded for the conversion to digital. Could the author please perhaps provide more detail on PCI in section 2. Instrument prototype setup and signal processing.

g.     What approaches can effectively enhance the capabilities of low-cost environmental sensors and should be included in the section: Calibration with machine learning techniques.

h.     What exactly is the Multi-Layer (ML) Pregressor neural network method, and what exactly is the objective of employing it?

i.       Which instrument enabled a direct comparison of the comparison measured instantaneous rain rate with the response of the study's sensor? The author also needs to clarify the instrument.

j.       The sensor interface's response to the impact of a falling drop is challenging to model, especially after the start of precipitation, when the moist layer dampens the impact of falling droplets. The author must explain what will happen.

Author Response

REPLY TO REVIEWER n.3

Dear reviewer,

Thank you very much for your comments concerning our manuscript entitled “Development and calibration of a low-cost, piezoelectric rainfall sensor through Machine Learning”. We have examined all your comments and suggestions carefully and made some revisions accordingly (please check the highlighted in blue in the new version). The following are our explanations for your questions or suggestions in your reviews.

R3Q1: Many typographical problems must be fixed before publication (th- to be represented in the superscript likewise fewer alphabet which needs to specify in the subscript also needs to be specified properly).

Answer1: The article has been improved further through a complete rereading, including the bibliography section where some problems about superscripts and subscripts existed.

R3Q2: The author has to provide a few lines about the disdrometer in the introductory section, and they also need to address the benefits and drawbacks of using a disdrometer.

Answer2: Some sentences about the disdrometer measurements, advantages and limits of their use have been included in the introduction section.

R3Q3: The author must include the application of the novel approach that is based on machine learning (ML).

Answer3: The novelty of the machine learning approach used in this work lies in the fact that most of the precipitation sensor calibration works are based on physical principles and models and on signal processing. In this case we used a mixed approach in which the quantities derived from a basic signal processing have been processed directly through machine learning algorithms to obtain the desired quantity. Some references of this approach applied to various types of sensors have been inserted in the text and are reported in the bibliography section.

A sentence has been rearranged in the introduction to specify that the novelty of the used approach is the application of Machine Learning methods for calibrating a precipitation sensor.

R3Q4:  Since commercially accessible non-catching devices (also called as disdrometers) that employ various technologies to gather not just rainfall rate but also Drop Size Distribution (DSD) have gained in popularity. The author need to discuss DSD in a few sentences?

Answer4: A short description of the DSD has been included in the introduction section when the disdrometers are introduced as requested also by Question2.

R3Q5: A low-cost piezoelectric rainfall sensor (PRS) was designed for the purpose of monitoring and detecting the instantaneous rainfall rate. This was done with the goal of achieving both cost efficiency and good gadget performance. Could the author provide much more details about the PRS?

Answer5: PRS is an acronym used to identify the sensor developed in this work. Probably from the previous version of the introduction it was not clear that the PRS is the object of the present work; therefore, it was written more clearly. Moreover, in the revised version of the manuscript the structure has been slightly changed and the description of the prototype of the PRS is provided in section 2 both in terms of implementation and in terms of signal and data processing.

R3Q6: The PCI (Peripheral Component Interconnect) card, where the analog signal is sampled, quantized, and coded for the conversion to digital. Could the author please perhaps provide more detail on PCI in section 2. Instrument prototype setup and signal processing.

Answer6: We apologize because the manuscript did not clearly explain that  the acquisition and processing system was a quite old PC, namely the Compaq D310 system. The integrated audio device integrated in the motherboard was used to acquire the signal provided by the piezoelectric transducer.

 The new version of the manuscript includes a description of the acquisition system in subsection2.1 “Instrument prototype setup and signal processing” by providing some technical details. 

R3Q7: What approaches can effectively enhance the capabilities of low-cost environmental sensors and should be included in the section: Calibration with machine learning techniques.

Answer7: In Okafor et. Al. a comparison between some selected machine learning methods as Simple Linear regression, Multiple Linear Regression, Artificial Neural Network, is provided for calibrating a low-cost air quality sensor. In this work a similar approach was adopted and the Section 2 describes both the implementation of the prototype and some tested calibration procedures.

Probably, the previous structure of the article was not clear enough. Now section 2 has been rewrote according to the Sensors journal formatting rules (Section 2 “Material and methods”) and in the introduction the organization of the various sections has been rewritten, to increase readability.

Okafor, N.U.; Alghorani, Y.; Delaney, D.T. , Improving data quality of low-cost IoT sensors in438
environmental monitoring networks using data fusion and machine learning approach. ICT439
Express 2020, 6, 3, 220-228. doi: 10.1016/j.icte.2020.06.004

R3Q8: What exactly is the Multi-Layer (ML) Pregressor neural network method, and what exactly is the objective of employing it?

Answer8: Sorry, there was a type error. The method was the “Multi-layer Perceptron Regression neural network”, sometime referred as Multi-layer PRegressor. It is included in the scikit-learn used for the development of this work (https://scikit-learn.org/stable/modules/generated/sklearn.neural_network.MLPRegressor.html#sklearn.neural_network.MLPRegressor).

It has been employed in this work as we decided to test a neural network algorithm and it has capability to learn non-linear models. We suppose in fact that the relationship between sensor measurements and precipitation was non-linear.

The type error has been corrected in the text. Thank you.   

R3Q9: Which instrument enabled a direct comparison of the comparison measured instantaneous rain rate with the response of the study's sensor? The author also needs to clarify the instrument.

Answer9: A sentence with some specifications about the measurement timing has been added to the manuscript in section 2.2. “Calibration with Machine learning methods”.  The sentence specifies that the PARSIVEL disdrometer, used as reference instrument, provides rain rate averaged on 1-minute acquisition time. The final measurements of our prototype are provided with the same time interval.

R3Q10: The sensor interface's response to the impact of a falling drop is challenging to model, especially after the start of precipitation, when the moist layer dampens the impact of falling droplets. The author must explain what will happen.

Answer10: A description of the observed effect has been added in Subsection 2.2 “Calibration with Machine learning methods”. The undesirable effects described in Subsection2.2 are the main causes of the failure of traditional calibration methods that suggested the introduction of machine learning methods.

Round 2

Reviewer 2 Report

Manuscript ID: sensors-1854422(v2)
Title: Development and calibration of a low-cost, piezoelectric rainfall sensor through Machine Learning
The authors have answered all the questions made in the first review of the manuscript. In my opinion, the manuscript may be accepted for publication in its present form. Only a very small comment is made. In line 88, the height of the sensor affects the meteorological parameters such as temperature or wind. It can be argued that does not affect rainfall. In line 90, where reads “meteorological parameters” should read “rainfall”.

Author Response

Dear reviewer,

you are right. In the text "meteorological parameter" has been replaced by "rainfall".

Thanks

Reviewer 3 Report

complete English correction is needed for throughout the manuscript. 

Author Response

Dear reviewer,

thank you for your comment.

English will be corrected by English department of Sensors Editorial office.

Best Regards,

Andrea Antonini